# Sources of bias in artificial intelligence that perpetuate healthcare disparities—A global review

Leo Anthony Celi[1,2,3], Jacqueline Cellini[4], Marie-Laure Charpignon[5], Edward Christopher Dee[6], Franck Dernoncourt[7], Rene Eber[8], William Greig Mitchell[9]*, Lama Moukheiber[10], Julian Schirmer[8], Julia Situ[11], Joseph Paguio[12], Joel Park[13], Judy Gichoya Wawira[14], Seth Yao[12], for MIT Critical Data

1 Massachusetts Institute of Technology, Institute for Medical Engineering and Science, Cambridge, MA, United States of America, 2 Harvard TH Chan School of Public Health, Department of Biostatistics, Boston, MA, United States of America, 3 Beth Israel Deaconess Medical Center, Department of Medicine, Boston, MA, United States of America, 4 Harvard Medical School, Department of Library Services, Boston, MA, United States of America, 5 Massachusetts Institute of Technology, Institute for Data, Systems and Society, Cambridge, MA, United States of America, 6 Harvard Medical School, Boston, MA, United States of America, 7 Adobe Inc, Adobe Research, San Jose, CA, United States of America, 8 Montpellier University, Montpellier Research in Management, Montpellier, France, 9 Harvard TH Chan School of Public Health, Boston, MA, United States of America, 10 Massachusetts General Hospital, Harvard Medical School, Boston, MA, United States of America, 11 Massachusetts Institute of Technology, Department of Computer Science and Molecular Biology, Cambridge, MA, United States of America, 12 Einstein Medical Center Philadelphia, Department of Medicine, Philadelphia, PA, United States of America, 13 BeiGene, Applied Innovation, Cambridge, MA, United States of America, 14 Emory University, Department of Radiology and Biomedical Informatics, Atlanta, GA, United States of America

* wmitchell@alumni.harvard.edu

**Data Availability Statement:** All the code used to build the machine learnings models is available at GitHub under https://github.com/Rebero/ml-disparities-mit.

## Abstract

### Background

While artificial intelligence (AI) offers possibilities of advanced clinical prediction and decision-making in healthcare, models trained on relatively homogeneous datasets, and populations poorly-representative of underlying diversity, limits generalisability and risks biased AI-based decisions. Here, we describe the landscape of AI in clinical medicine to delineate population and data-source disparities.

### Methods

We performed a scoping review of clinical papers published in PubMed in 2019 using AI techniques. We assessed differences in dataset country source, clinical specialty, and author nationality, sex, and expertise. A manually tagged subsample of PubMed articles was used to train a model, leveraging transfer-learning techniques (building upon an existing BioBERT model) to predict eligibility for inclusion (original, human, clinical AI literature). Of all eligible articles, database country source and clinical specialty were manually labelled. A BioBERT-based model predicted first/last author expertise. Author nationality was determined using corresponding affiliated institution information using Entrez Direct. And first/last author sex was evaluated using the Gendarize.io API.

**Funding:** LAC is funded by NIBIB grant R01 EB017205. The funders of the grant had no role in study design, data collection and analysis, decision to publish, or preparation of the manuscript. Grant detail can be found at: https://grantome.com/grant/NIH/R01-EB017205-01A1 No other authors received any specific funding for this work.

**Competing interests:** Leo Anthony Celi is the Editor-in Chief of PLOS Digital Health and Judy Gichoya Wawira is a Section Editor for PLOS Digital Health.

## Results

Our search yielded 30,576 articles, of which 7,314 (23.9%) were eligible for further analysis. Most databases came from the US (40.8%) and China (13.7%). Radiology was the most represented clinical specialty (40.4%), followed by pathology (9.1%). Authors were primarily from either China (24.0%) or the US (18.4%). First and last authors were predominately data experts (i.e., statisticians) (59.6% and 53.9% respectively) rather than clinicians. And the majority of first/last authors were male (74.1%).

## Interpretation

U.S. and Chinese datasets and authors were disproportionately overrepresented in clinical AI, and almost all of the top 10 databases and author nationalities were from high income countries (HICs). AI techniques were most commonly employed for image-rich specialties, and authors were predominantly male, with non-clinical backgrounds. Development of technological infrastructure in data-poor regions, and diligence in external validation and model re-calibration prior to clinical implementation in the short-term, are crucial in ensuring clinical AI is meaningful for broader populations, and to avoid perpetuating global health inequity.

## Author summary

Artificial Intelligence (AI) creates opportunities for accurate, objective and immediate decision support in healthcare with little expert input–especially valuable in resource-poor settings where there is shortage of specialist care. Given that AI poorly generalises to cohorts outside those whose data was used to train and validate the algorithms, populations in data-rich regions stand to benefit substantially more vs data-poor regions, entrenching existing healthcare disparities. Here, we show that more than half of the datasets used for clinical AI originate from either the US or China. In addition, the U.S. and China contribute over 40% of the authors of the publications. While the models may perform on-par/better than clinician decision-making in the well-represented regions, benefits elsewhere are not guaranteed. Further, we show discrepancies in gender and specialty representation–notably that almost three-quarters of the coveted first/senior authorship positions were held by men, and radiology accounted for 40% of all clinical AI manuscripts. We emphasize that building equitable sociodemographic representation in data repositories, in author nationality, gender and expertise, and in clinical specialties is crucial in ameliorating health inequities.

## Introduction

While there is no exact definition for artificial intelligence (AI), AI broadly refers to technologies that allow computer systems to perform tasks that would normally require human intelligence [1–3]. Machine learning (ML), deep learning (DL), convolutional neural networks (CNN), and natural language processing (NLP) are forms of automated decision-making techniques that exist on the AI continuum, each requiring varying degrees of human supervision. These forms of AI have all been rigorously applied to healthcare in both clinical and academic settings, with particular acceleration in the last decade [4–10]. In a world where clinical decisions are often impacted by conjecture, tradition, convenience, and habit, AI offers the

possibility of robust and consistent decision-making–in many cases performing on par with or better than human physicians, [11–13] creating a world wherein clinical practice is routinely aided by AI [14].

The recent proliferation of AI in healthcare has been facilitated by factors such as increased cloud storage (e.g., mass data compilation, labelling, and retrieval) and enhanced computer power and speed, creating possibilities for reduced clinical errors and semi-automated outcome prediction and enabling patients to promote their own health with their own unique data [8,9,12]. However, the introduction of AI into healthcare comes with its own biases and disparities; it risks thrusting the world toward an exaggerated state of healthcare inequity [15–23]. Repeatedly feeding models with relatively homogeneous data, suffering from a lack of diversity in terms of underlying patient populations and often curated from restricted clinical settings, can severely limit the generalisability of results and yield biased AI-based decisions [24]. For example, there is no guarantee that a model predicting diabetic retinopathy (DR) built using the clinical trial data of a relatively small, homogeneous urban population within the U.S. would be applicable to a cohort of patients living in rural areas of Japan [25]. Unequal access to the very factors that have facilitated the proliferation of AI in healthcare (e.g., readily available electronic health information and computer power) may be widening existing healthcare disparities and perpetuating inequities in who benefits most from such technological progress.

Unless AI represents countries and clinical specialties equally in healthcare, or unless models are at the very least externally validated on diverse patient populations, entrenched disparities in healthcare may persist. Technological advancement alone will not drive the world toward a state of healthcare equity; it *must* be coupled with an understanding of the under- and over-represented patients in healthcare ML and with international efforts to combat the risk of AI bias.

The current study describes the landscape of AI in clinical medicine to better understand the disparities in data sources and patient populations.

## Methods

### Original search strategy and selection criteria

The present study applies ML techniques to comprehensively review all medical and surgical (hereafter referred to as clinical) manuscripts published in PubMed through 2019 which employ AI techniques (defined here as either AI, ML, DL, NLP, computer vision [CV], or CNN). After identifying all such clinical manuscripts, we describe differences in both populations captured in databases and clinical specialty and in authors' nationality, gender, and domain of expertise to elucidate the extent of the disparities affecting AI in healthcare.

Our review can best be defined as a *scoping review*. Scoping reviews are a novel approach well-suited for describing the mix of literature in a given area, in terms of the clinical specialties represented and the types of research questions being addressed [26–30]. Such types of research topics include diagnosis, causation, and prognosis. Whereas a systematic review aims to answer a specific clinical question, using a rigid protocol determined *a priori* (including an assessment of research quality and risk of bias), a scoping review maps a body of literature, identifies trends and deficiencies, and addresses broader research questions such as areas to prioritize for research.

### Eligibility of PubMed articles

Articles were presently deemed eligible for inclusion if they were (i) original research (i.e., not post-hoc analyses, to avoid double-counting); (ii) medical or surgical research (i.e., not

viewpoints, commentaries, or research without clinically applicable findings); (iii) human research (e.g., studies that assessed mouse neural networks were excluded); (iv) published in English; and (v) employing either AI, ML, DL, NLP, CV, or NN techniques.

## Search methods

The PubMed search system allows a narrowed literature search using medical subject headings (MeSH) and controlled vocabulary, filtering for specific terms and phrases in the titles of articles (ti) within PubMed [31]. The following search was undertaken for the present study: ("machine learning"[Majr] OR ("machine"[ti] AND "learning"[ti]) OR "machine learning"[ti] OR "AI"[ti] OR "Artificial Intelligence"[ti] OR "artificially intelligent"[ti] OR "Artificial Intelligence"[MeSH] OR "Algorithms"[MeSH] OR "algorithm*"[ti] OR "deep learning"[ti] OR "computer vision"[ti] OR "natural language processing"[ti] OR "neural network*"[ti] OR "neural networks, compu-ter"[MeSH] OR "intelligent machine*"[ti]) AND 2019/1/1:2019/12/31[Date—Publication].

## Machine learning model building to predict shortlist of eligible articles

The PubMed search above identified 30,576 articles published in 2019. To build our training dataset, we randomly selected 2,000 articles and manually screened them for eligibility using Covidence review software [32]. Two independent reviewers assessed the 2,000 articles for eligibility by checking (i) the manuscript title, (ii) the abstract, and (iii) the full text (in cases where eligibility was still unclear). Where the two reviewers disagreed about an article's eligibility, both manually cross-checked the full article again; a third reviewer's decision determined the final eligibility.

Articles in PubMed missing either a title or an abstract were excluded from analyses, since there would not be sufficient information to characterize the machine learning model(s) being described in the paper. *A-posteriori* manual review of the full texts revealed that such articles were almost exclusively commentaries, editorials, letters, and post-hoc analyses rather than full research manuscripts, confirming their ineligibility to be part of our scoping review.

Using the subset of 2,000 labelled manuscripts classified into two categories (eligible and non-eligible), a machine learning model was trained to predict eligibility for the remaining set of 2019 PubMed search results for clinical artificial intelligence articles according to the above-mentioned criteria. The trained model predicted the remaining articles to be either eligible or non-eligible. Our model leveraged transfer-learning techniques building upon BioBERT, a biomedical language representation model designed for biomedical text mining tasks [33]. BioBERT was originally trained on different combinations of the following text corpora: English Wikipedia (Wiki), BooksCorpus (Books), and PubMed abstracts (PubMed). The pre-trained BioBERT model was sourced from Huggingface. We used Biobert_v1.1_pubmed by Monologg [34]. First, we removed the final layer of BioBERT and replaced it with a final classification layer tailored to our supervised binary classification task. The final layer was fine-tuned using titles and abstracts from 1,600 of the 2,000 manually screened articles, saving 20% for validation. Next, the full model was fine-tuned on the 1,600 articles. Lastly, we validated the model on the remaining 400 manually screened articles. Two other independent reviewers checked the predictions output by the model, controlling for potential biases and outliers among the resulting probability scores on a scale from 0 to 100%. The code for this model is available at Github (https://github.com/Rebero/ml-disparities-mit).

## Extracting the source of each paper's database and its clinical specialty

We determined the country from which the database was sourced by manually tagging two randomly generated subsamples of 300 papers each from the shortlist of 7,314 articles labelled

**Table 1. Tagged subsamples of distribution of database nationality in AI in medicine.**

| Subsample 1 | | | Subsample 2 | | |
|---|---|---|---|---|---|
| Country | Count | Percent | Country | Count | Percent |
| United States of America | 124 | 48.8 | United States of America | 82 | 32.7 |
| China | 30 | 11.8 | China | 39 | 15.5 |
| United Kingdom | 18 | 7.1 | Germany | 19 | 7.6 |
| Germany | 10 | 3.9 | United Kingdom | 16 | 6.4 |
| Australia | 8 | 3.1 | South Korea | 10 | 4.0 |
| Japan | 8 | 3.1 | Canada | 10 | 4.0 |
| Canada | 7 | 2.8 | Netherlands | 9 | 3.6 |
| South Korea | 5 | 2.0 | Japan | 7 | 2.8 |
| Netherlands | 5 | 2.0 | France | 5 | 2.0 |
| Austria | 4 | 1.6 | Spain | 5 | 2.0 |
| France | 4 | 1.6 | Australia | 4 | 1.6 |
| North Korea | 4 | 1.6 | Turkey | 4 | 1.6 |
| Brazil | 3 | 1.2 | Switzerland | 4 | 1.6 |
| Israel | 3 | 1.2 | India | 4 | 1.6 |
| Italy | 3 | 1.2 | Italy | 3 | 1.2 |
| Spain | 2 | 0.8 | Taiwan | 3 | 1.2 |
| Switzerland | 2 | 0.8 | Austria | 3 | 1.2 |
| Sweden | 2 | 0.8 | Denmark | 3 | 1.2 |
| Czechia | 2 | 0.8 | Mexico | 3 | 1.2 |
| India | 1 | 0.4 | New Zealand | 2 | 0.8 |
| Bangladesh | 1 | 0.4 | Czechia | 2 | 0.8 |
| Thailand | 1 | 0.4 | Norway | 2 | 0.8 |
| Zambia | 1 | 0.4 | Sweden | 2 | 0.8 |
| Slovenia | 1 | 0.4 | Malaysia | 1 | 0.4 |
| Malaysia | 1 | 0.4 | Indonesia | 1 | 0.4 |
| Mexico | 1 | 0.4 | Finland | 1 | 0.4 |
| Taiwan | 1 | 0.4 | Serbia | 1 | 0.4 |
| New Zealand | 1 | 0.4 | Brazil | 1 | 0.4 |
| Pakistan | 1 | 0.4 | Israel | 1 | 0.4 |
| - | - | - | South Africa | 1 | 0.4 |
| - | - | - | Portugal | 1 | 0.4 |
| - | - | - | Belgium | 1 | 0.4 |
| - | - | - | Iran | 1 | 0.4 |
| **Total** | **254** | **100** | **Total** | **251** | **100** |

as eligible. To properly label each instance, we independently reviewed the text (usually the methodology section or the data availability statement) or the supplementary material. In instances where the provenance of the database used to build the model(s) was not declared in either the manuscript or the supplementary material, or when it was imprecise (e.g., "multiple countries throughout Eastern Europe"), the label was marked as missing. Upon completion of this task, the two reviewers compared their two manually labelled subsamples for consistency. The two subsamples were then combined into a sample of 600 manually curated papers. Table 1 shows the results of the manual labelling of each of the 300 subsamples, while Fig 1A and 1B show the combined results.

In parallel, the clinical specialty of the manuscript was similarly determined by two independent reviewers manually curating two subsamples of 300 papers that were randomly

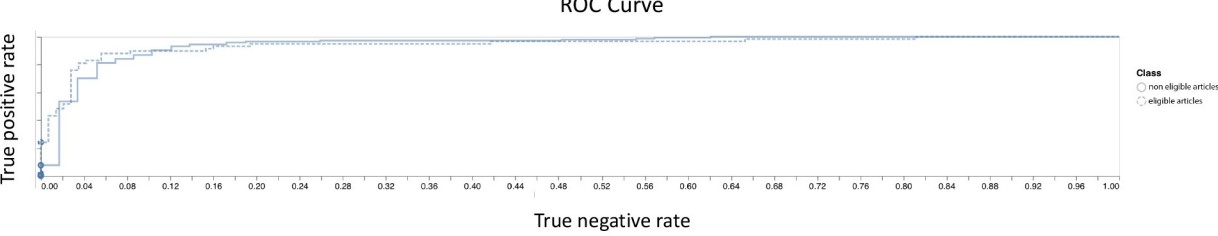

(a) – confusion matrix and ROC curve (a)

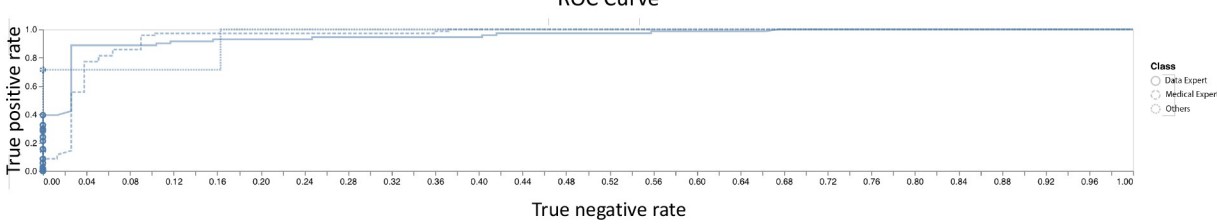

(b) - confusion matrix and ROC curve (b)

**Fig 1.** (a) Confusion matrix and ROC curve. (b) Confusion matrix and ROC curve.

selected from the shortlist of articles labelled as eligible. The clinical specialty of each paper was determined by assessing (i) which medical or surgical specialist would use the AI findings in their own clinical practice, (ii) which medical or surgical specialty the AI model was being compared with, or (iii) the specialty of the journal in which the article was published, in cases where neither (i) nor (ii) was clear. For each of the two subsamples, the two reviewers compared their labels; when needed, a third reviewer's decision determined the final specialty of the paper. If multiple specialties were deemed related, then multiple labels were assigned. Conversely, if no specialty was deemed relevant, no label was assigned. Similar to the process used to label the source of the database, the two subsamples of 300 papers were then combined to present the final results. The results of the manual labelling of each of the 300 subsamples can

**Table 2. Tagged subsamples of distribution of paper specialty in AI in medicine.**

| Subsample 1 | | | Subsample 2 | | |
|---|---|---|---|---|---|
| Specialty | Count | Percent | Specialty | Count | Percent |
| Radiology | 147 | 48.4 | Radiology | 71 | 30.1 |
| Pathology | 35 | 11.5 | Neurology | 29 | 12.3 |
| Ophthalmology | 24 | 7.9 | Medicine | 28 | 11.9 |
| Cardiology | 15 | 4.9 | Ophthalmology | 16 | 6.8 |
| Oncology | 14 | 4.6 | Cardiology | 14 | 5.9 |
| Neurology | 11 | 3.6 | Pathology | 14 | 5.9 |
| Surgery | 11 | 3.6 | Psychiatry | 13 | 5.5 |
| Pediatrics | 6 | 2.0 | Oncology | 10 | 4.2 |
| Dermatology | 6 | 2.0 | Intensive Care | 10 | 4.2 |
| Gastroenterology | 5 | 1.6 | Orthopedics | 6 | 2.5 |
| Anesthesia | 4 | 1.3 | Gastroenterology | 5 | 2.1 |
| Intensive Care | 3 | 1.0 | Public Health | 5 | 2.1 |
| Emergency | 3 | 1.0 | Respiratory | 4 | 1.7 |
| Obstetrics/Gynecology | 3 | 1.0 | Dermatology | 3 | 1.3 |
| Public Health | 3 | 1.0 | Pediatrics | 2 | 0.8 |
| Hematology | 2 | 0.7 | Surgery | 1 | 0.4 |
| Endocrinology | 2 | 0.7 | Obstetrics | 1 | 0.4 |
| Administration | 1 | 0.3 | Nephrology | 1 | 0.4 |
| Epidemiology | 1 | 0.3 | Maxillofacial surgery | 1 | 0.4 |
| Urology | 1 | 0.3 | Infectious Diseases | 1 | 0.4 |
| Rheumatology | 1 | 0.3 | Dentistry | 1 | 0.4 |
| Orthopedics | 1 | 0.3 | - | - | - |
| Histology | 1 | 0.3 | - | - | - |
| Genetics | 1 | 0.3 | - | - | - |
| Respiratory | 1 | 0.3 | - | - | - |
| Psychiatry | 1 | 0.3 | - | - | - |
| Infectious Diseases | 1 | 0.3 | - | - | - |
| **Total** | **304** | **100** | **Total** | **236** | **100** |

be found in Table 2, and the overall combined results in Fig 2. A heat map was generated using the geopandas package, an open source project to work and plot geospatial data in Python (Fig 2B) [35].

## Machine learning model to predict each author's domain of expertise

The domain of expertise of each author was determined using a similar approach to the identification of papers' clinical specialties. Specifically, a machine learning model was trained to predict the domain of expertise of the first and last authors for the entire shortlist of eligible articles. First, the domains of expertise of the first and last authors were manually labelled in the subset of eligible articles from the randomly generated 2,000 articles originally screened in Covidence. For this, two independent reviewers classified the first and last author as either (i) a data expert (i.e., statistician), (ii) a domain expert (i.e., a medical specialist or surgeon), or (iii) other. Results of the manually labelled subsamples were compared between reviewers to ensure consistency; a third reviewer made the final decision. Second, we trained a classifier to predict the expertise of the first and last author among the three abovementioned categories. For this, we used transfer-learning techniques and fine-tuned an existing BioBERT model. The

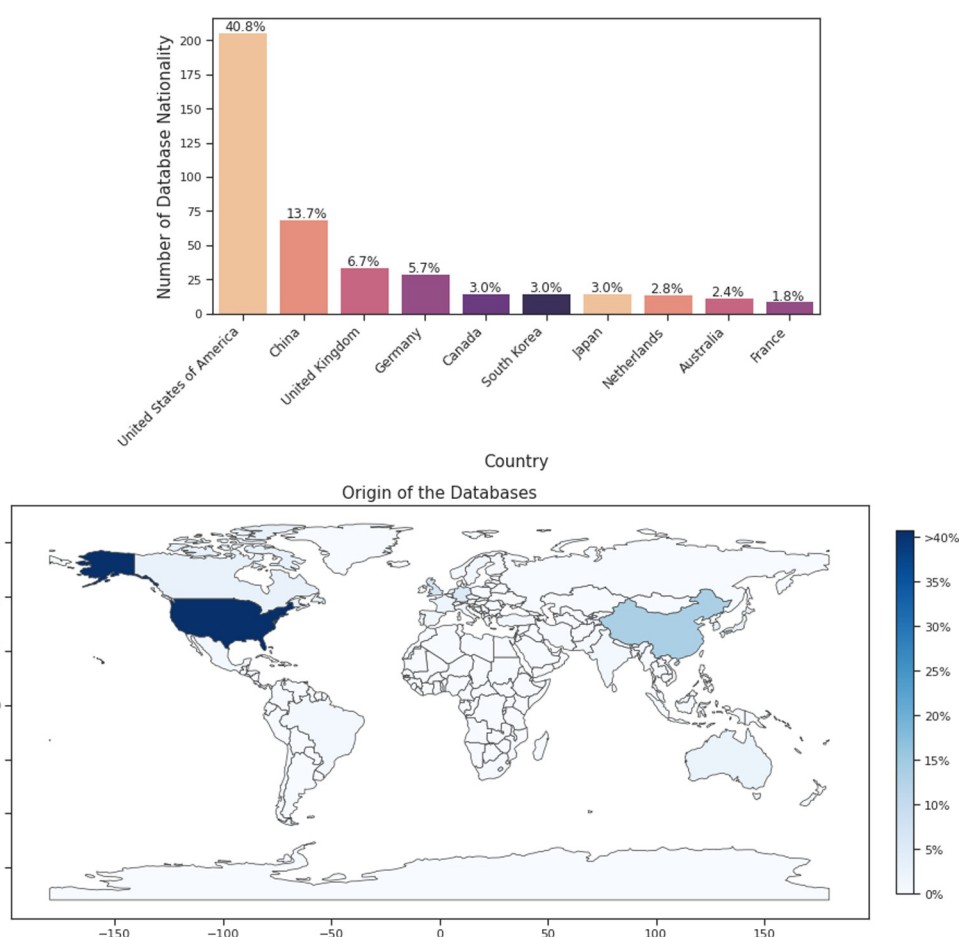

**Fig 2.** (a) Distribution of overall database nationality in AI in medicine. (b) Heatmap of distribution of overall database nationality in AI in medicine (reference #35)

model was trained on the 1,852 (i.e., 92.6%) manually screened articles containing the considered author's affiliation as well as their background, following the same process than for the model built to predict the eligibility of articles. Next, the full model was fine-tuned. Lastly, we validated the model on the remaining 148 (i.e., 7.4%) manually screened articles. The corresponding code is available at Github (https://github.com/Rebero/ml-disparities-mit).

## Approach to identify each author's nationality and gender

For each publication, the list of authors and their corresponding affiliated institutions were acquired using Entrez Direct (EDirect), the suite of interconnected databases made available by the National Center for Biotechnology Information (NCBI) and accessible via a Unix terminal window [36]. Given that the raw data contained the country of each research institution present in the list, the field was first tokenized and the resulting array of tokens was then parsed to retrieve country names. Each author was subsequently linked to the country in which their institution is based. Occasionally, some authors were affiliated with institutions spanning two or more countries. In those instances, all of their affiliated countries were mapped to the corresponding author.

The first name of all authors was extracted from each article using metadata and subsequently processed through the Genderize.io Application Programming Interface (API) [37–

[39]. The API contains a collection of previously annotated first names and their reported gender. Based on the ratio of male to female examples stored in the underlying database, the API calculates the probability of a male or female gender and assigns the most likely gender to the first name under consideration.

## Results

### Final shortlist of eligible clinical AI articles published in PubMed through 2019

The abovementioned PubMed search yielded 30,576 clinical AI articles published through 2019 prior to further screening for eligibility. Of the 2,000 articles manually screened, 368 (i.e., 18.4%) were found to be eligible. This labelled subset was subsequently used to train and test the model used to predict article eligibility. Our model achieved both a high Matthews correlation coefficient (MCC) of 0.88 and a high Area under the ROC Curve (AUROC) of 0.96 on the test dataset. In the case of a tie (i.e., a predicted probability for eligibility of 50%), we biased the model to err on the side of classifying an article as eligible rather than ineligible. The objective was indeed to limit the number of articles that could be misclassified as ineligible while they should in fact be included (Fig 1A). Using such a framework, we ran our ML model to assess eligibility of all articles published in 2019. A total of 7,314 (i.e., 23.9%) were deemed "eligible", based on the criteria described above.

### Identification of the sources of the database(s) being used in the papers

Of the two subsamples of 300 eligible papers each that we manually labelled, 254 and 251 (i.e., ~84%) mentioned the sources of the database(s) being used, either in the manuscript or in the supplementary material (Table 1). The U.S. accounted for the majority of the data sources in both subsamples (48.8% and 32.7%, respectively), followed by China (11.8% and 15.5%, respectively). Other countries represented in both subsets, including the U.K., Germany, Canada, and South Korea, had a substantially smaller prevalence (Table 1). Results emanating from the two labelled subsets were thus deemed comparable and combined; pooled results are presented in Fig 2A and 2B. In sum, the U.S. contributed the vast majority of AI datasets published in 2019 (40.8%), followed by China (13.7%), the U.K. (6.7%), and Germany (5.7%).

### Determination of each paper's clinical specialty

Radiology was the most represented clinical specialty in both subsamples of manually labelled data (48.4% and 30.1%, respectively) (Table 2). Pathology, ophthalmology, neurology, and cardiology comprised four of the following five specialties in both subsets. Results were subsequently concluded to be comparable and combined (Table 2). Radiology brought the highest number of clinical AI studies in 2019 (accounting for 40.4% of pooled studies), followed by pathology (9.1%), neurology, ophthalmology (both 7.4%), cardiology (5.4%), and internal/hospital/general medicine (5.2%) (Fig 3).

### Estimation of each author's nationality, domain of expertise, and gender

Overall, nationalities of 123,815 authors were extracted from the 7,314 eligible articles and distribution of author nationalities are shown in Fig 3. Most authors came from either China or the U.S. (24.0% and 18.4%, respectively), followed by Germany (6.5%), Japan (4.3%), the U.K. (4.1%), Italy, France, Canada (each 3.6%), Australia, the Netherlands (each 2.8%), Spain (2.7%), and India (2.3%) (Fig 4). The model for predicting the author background achieved

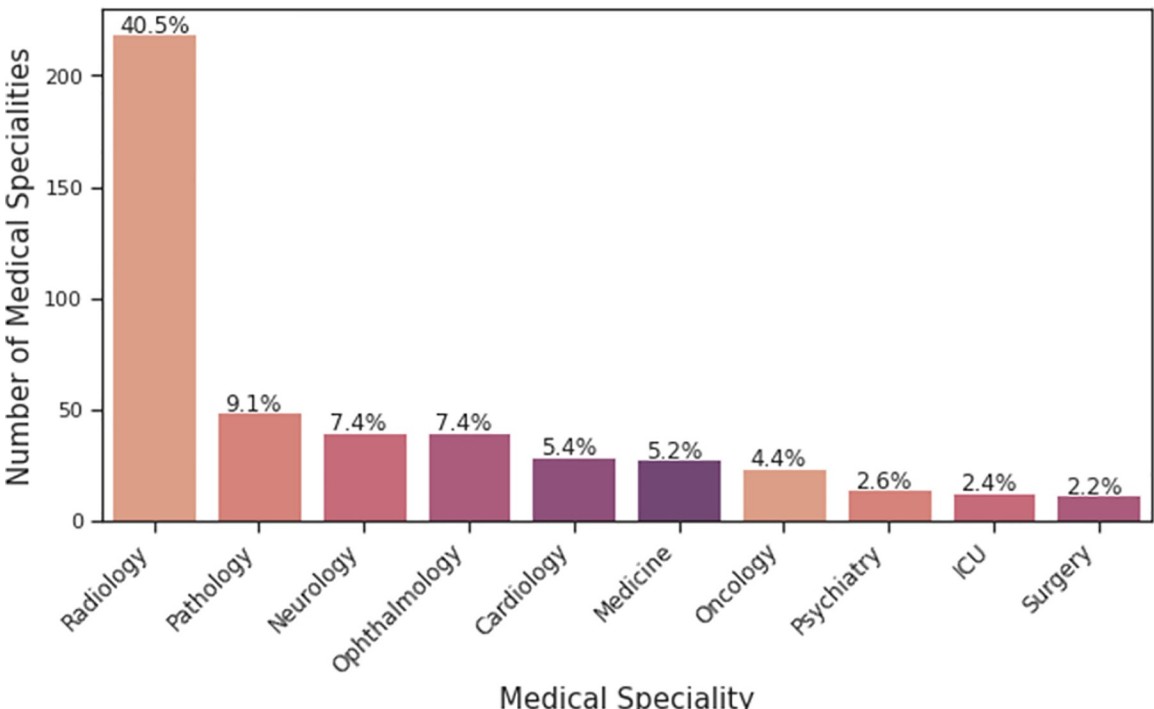

**Fig 3. Distribution of overall paper specialty in AI in medicine.**

both a high Matthews correlation coefficient (MCC) of 0.85 and a high Area under the ROC Curve (AUROC) on the test dataset (Fig 1B).

We found that both the first and last authors were predominantly data experts (59.6% and 53.9%, respectively), followed by domain experts (36.5% and 41.4%, respectively), and other experts (3.9% and 3.8%, respectively) (Fig 5). Overall, the majority of first and last authors were male (combined: 74.1%, first: 73.6%, last: 79.5%) (Fig 6).

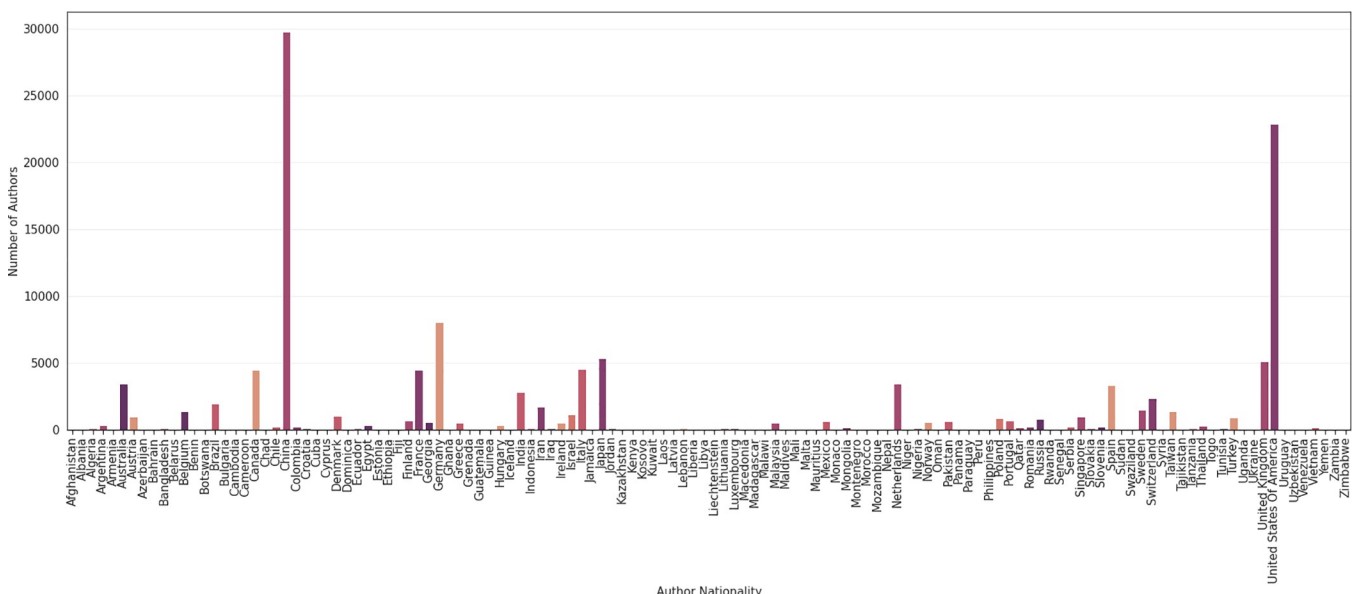

**Fig 4. Distribution of overall author nationality in AI in medicine.**

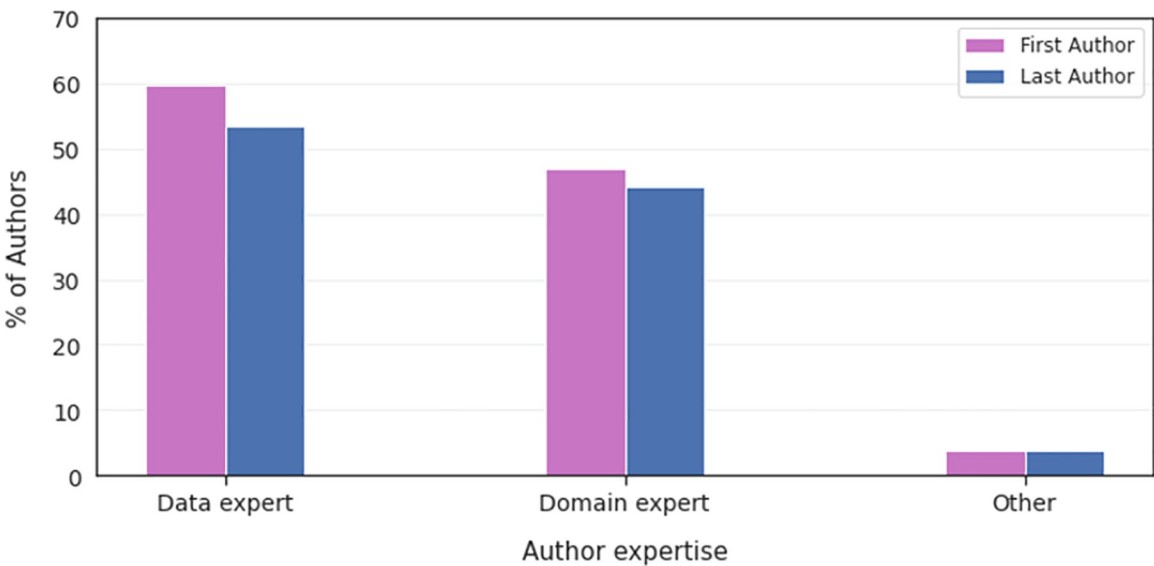

**Fig 5. Distribution of first and last author expertise in AI in medicine.**

## Discussion

Our study demonstrates substantial disparities in the data sources used to develop clinical AI models, representation of specialties, and in authors' gender, nationality, and expertise. We found that the top 10 databases and author nationalities were affiliated with high income countries (excluding China), and over half of the databases used to train models came from either the U.S. or China [40]. While pathology, neurology, ophthalmology, cardiology, and internal medicine were all similarly represented, radiology was substantially overrepresented, accounting for over 40% of papers published in 2019, perhaps due to facilitated access to image data.

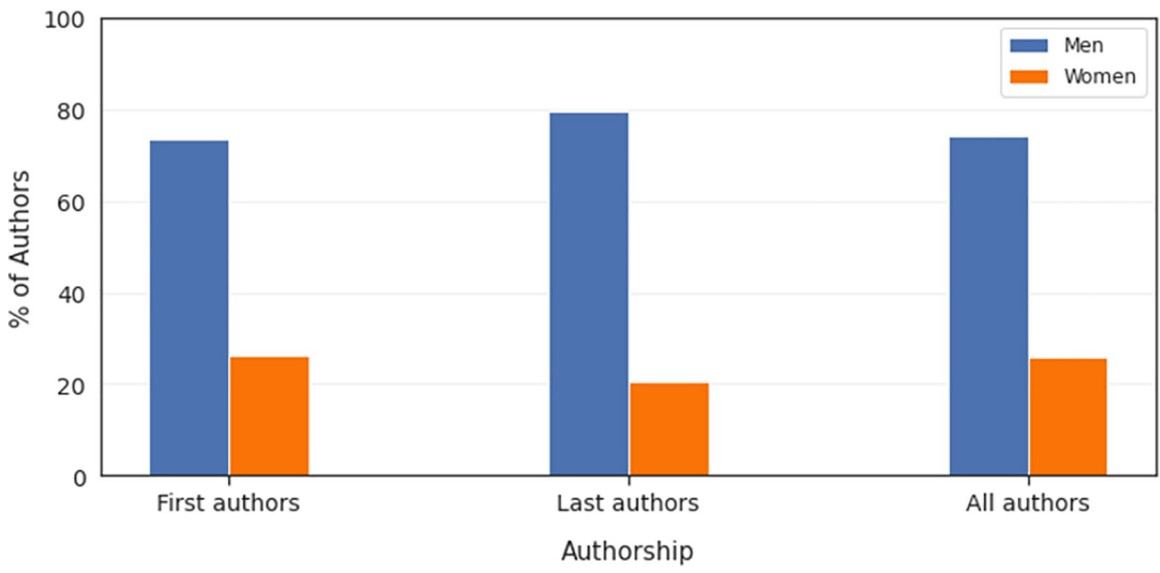

**Fig 6. Distribution of overall author gender in AI in medicine.**

Additionally, more than half of the authors contributing to these clinical AI manuscripts had non-clinical backgrounds and were three times more likely to be male than female.

Here, we show that most models are trained with U.S. and Chinese data. This finding is unsurprising, given the importance of cloud storage as well as computer power and speed in gathering clinical data to build models and the relatively advanced technological and AI infrastructure within these countries. It intuitively follows that most authors of clinical papers implementing AI methods are from these regions, too. We stress that while such disproportionate overrepresentation of U.S. and Chinese patient data in AI brings the risk of bias and disparity, it does not negate/undercut/undermine the value of understanding when and why these models remain beneficial, particularly in data-poor and demographically diverse regions (both within these nations and globally).

Model pre-training offers a particular mechanism through which AI built in data-rich areas might be applied in data-poor regions. Pre-training has grown primarily through NLP, where contextual word embedding models such as Word2Vec, [41] GloVe, [42] and BERT have improved performance of models across a variety of tasks [43]. Pre-trained models are generally trained in one task and subsequently leveraged to solve similar problems. For example, BERT models were built using clinical notes from the publicly available MIMIC-III database for enhanced disease prediction, [44] and have since been used to improve clinical Japanese data representation written in other languages, [45] and integrated with other complementary medical knowledge databases [46]. While development costs can be high (e.g., training GPT-3, a novel large-scale language, cost over $10 million and required more than 300GB of memory), [47] such frameworks negate the limitations of working with smaller datasets through leveraging existing models for recalibration for diverse target populations. Understanding how and why primary models are applicable to different cohorts (i.e., with previously unseen data) also contributes towards refining initial model methodology and proposing robust model alternatives.

The lack of diverse digital datasets for ML algorithms can amplify systematic underrepresentation of certain populations, posing a real risk of AI bias. This bias could worsen minority marginalisation and widen the chasm of healthcare inequality [10]. Although understanding these risks is crucial, to date inadequate appreciation for them has occurred. A recent review of international COVID-19 datasets used to train ML models for predicting diagnosis and outcomes showed that of the 62 shortlisted manuscripts, *half* neither reported sociodemographic details of the data used to train their models nor made any attempt to externally validate their results or assess the sensitivity of the models. None of the studies performed a proper assessment of model bias [48].

Moreover, applying non-validated models (curated with homogeneous data) on sociodemographically diverse populations poses a considerable risk [49]. For example, if a model created with exclusively U.S. data were used to predict the mortality of a Vietnamese COVID-19 population (without external validation), predictions might be inaccurate, given the model's founding "understanding" of outcomes formulated from an exclusively U.S. population. If the physician clinically applying model predictions did not fully appreciate this risk, the model might hold undue (because unsubstantiated) influence over the decision to escalate or withdraw care. Alternatively, if this limitation *is* known, the model may simply not be used at all, thereby restricting benefits to the U.S. population upon whose data it was exclusively trained. Either outcome is disadvantageous to populations not represented in the large datasets commonly used to build these models.

Increased computer power and massive dataset availability has driven the growth of ML in healthcare, especially over the last decade. The volume of data being generated cannot be understated—$10^8$ bytes in the U.S. alone, with annual growth of 48%—as well as increasing

uptake of digital electronic healthcare systems and proliferation of medical devices [50]. The ability of machines to interpret and manipulate data relatively-independent of human supervision/input brings with it the immense potential for improvements in healthcare quality and accessibility–particularly in resource-poor settings where specialist opinion is not as readily available [51,52].

While we echo the call for increased data diversification and equity to remedy disparities in data representation, such worthwhile efforts will take time and require the development of complex and costly technological infrastructure. External validation may provide a much more practical, short-term solution to healthcare inequities born of data disparity. Investing in the infrastructure for local validation and model re-calibration will also lay the groundwork for eventual contribution of local data to international data repositories.

The importance of externally validating models *within* populations should also be noted. Indeed, this was highlighted in a recent study that used CNNs to assess pneumonia severity in over 150,000 chest radiographs in the U.S. [53] Despite high model accuracy within the centre in which it was built, authors found that the model performed poorly when applied to data from another institution *within* the same country. Similar within-nation biases have been shown in recent ophthalmic AI models predicting DR with similar or greater accuracy than fully-trained ophthalmologists.[13,25] Closer inspection reveals the models to be built using data originally from the RISE and RIDE trials (composed of ~530 North/South American patients, of whom >80% were white and <1% Native American/Alaskan) [54]. While model predictions may be accurate and applicable for white North Americans, if they were used for a Native American population (who suffers more than double the prevalence of type II diabetes, as high as 49%), their accuracy would be limited and potentially misleading [25,52,55,56]. Benefits of *international* external validation are ever-apparent, with 172 countries (totaling ~3.5 billion people) lacking any public ophthalmic data repository); [57,58] however, the importance of *within*-nation validation cannot be understated.

The above examples outline the complexities of the concept of generalizability itself. Because the effects of hospital- and physician-level decisions on model performance cannot be neglected, models trained under specific contexts and assumptions should be assessed for performance when applied elsewhere (even within the same country) and continuously monitored. Only then will we begin to understand the clinical circumstances in which AI-based models provide the most insight and to identify settings in which they do not. Such learnings will in turn help increase the utility of clinical AI models [24].

Unsurprisingly, we found that clinical specialties with higher volumes of stored imaging were disproportionately overrepresented. On the one hand, this disparity underscores that the benefits of AI advances may not apply universally across medical and surgical specialties. As benefits of AI in medicine become clearer, efforts will be needed to train AI experts in various fields, especially those for which key areas that could be augmented by AI have yet to be developed. On the other hand, our findings also suggest a means through which disparities among clinical specialties could potentially be mitigated: the role that radiology, pathology, ophthalmology, neurology, and cardiology have played in leading the development of clinical AI may yield efficient computational methods that are either applicable or transferrable to other fields. For example, developments pioneered largely by radiology are now being applied to radiation oncology [59–61]. Furthermore, existing disparities in ML in medicine may galvanize future efforts in other clinical fields to work towards inclusivity in the data being collected and towards equity in the deployment of models based upon them.

As more AI/ML-based software becomes widely available for screening and diagnosis (e.g., breast cancer and melanoma screening tools, COVID-19 prognostic models, etc.), regulations surrounding their approval must also evolve [3]. Currently, such medical devices can be

approved through various regulatory pathways. The 510(k) pathway, which requires the proof of "substantial equivalence" to an already-marketed device (i.e., pre-existing AI/ML devices), [62] has been one of the primary approval pathways for such medical devices in the U.S. To date, the safety of medical devices cleared through this pathway has been controversial. As more medical devices employing AI/ML techniques are approved, an understanding of the populations they are recommended for–and how they differ from those who contributed the data used to train the underlying models–is warranted.

Our analysis also revealed that the first and last authors of clinical AI papers published in 2019 were three times more likely to be men than women and marginally more likely to be data specialists rather than clinical experts. The underrepresentation of women in research, especially women of color, and in datasets used to inform AI is well-documented and discussed in detail elsewhere [63,64]. Historically, algorithms have only poorly accounted for differences in patient characteristics like biological sex, although individual-level factors are often involved in complex causal relationship patterns. For example, biological differences between patients can affect the metabolism and efficacy of certain pharmacological compounds and should be accounted for when predicting outcomes of treatment with a certain drug [63]. Gender discrimination in clinical AI authorship is even less studied and not comprehensively described. While a number of metrics based on funding, hiring, grant allocation, and publication statistics suggests that gender disparities in healthcare research may have decreased in the past few years, subtler disparities persist. Indeed, recent research has shown that women are still significantly underrepresented in the prestigious first and last author positions and in single-authored papers. Our results, although specific to the field of clinical AI, are consistent with this literature [65]. It is concerning that such disparities in clinical AI research and publishing exist, despite a higher proportion of bachelor degrees being awarded to women since the 1980s and gender equity among PhD recipients in the U.S. [66] Further, it remains to elucidate the implications of the lack of gender balance in clinical AI research authorship for patient outcomes.

## Limitations

Nonetheless, our study presents several methodological limitations. As a scoping review, our analysis includes data from papers published in 2019 alone. Although a great number of studies were included, we were unable to systematically assess temporal trends in clinical AI disparities. We measure these disparities by quantifying the lack of international representation in terms of both data sources and research authors but do not account for disparities that exist within countries. Future work must determine how AI/ML disparities might manifest locally and globally and identify ways to mitigate them, taking into consideration between- and within-country population differences. Moreover, classification by clinical specialty may be restrictive and not holistically capture the many fields in which model-derived findings can apply. For example, an ML-based study of chest radiographs would augment a radiologist's work but might also impact the care provided by a cardiologist, a pulmonologist, an internal medicine physician, and many other clinicians to their patients. Importantly, the gender inference methods used in this paper, as with any methods outside of direct survey and self-identification, are imperfect and likely to have misgendered a subset of the author population. Additionally, the Genderize.io API was unable to assign a gender to every author name. Further, the API is not inclusive beyond the gender binary (e.g., intersectional gender and they/ them pronouns are not considered). We appreciate that author affiliation is not necessarily representative of author nationality, as it was used here to describe. Indeed, despite our own authors hailing from Australia, Canada, the U.S., Philippines, France and Germany–all of our

affiliations suggest we are from the U.S.–highlighting further intrinsic biases in data, subsequently embedded into AI if not carefully accounted for. Of note, only articles published in English were included; while still capturing the majority of international clinical (i.e., medical and surgical) journals, this design choice may have led to selection bias. Finally, because the ML models we built were trained to run solely on article abstracts, the exact data sources–often mentioned in the supplementary material and not easily accessible–could only be extracted for a portion of the articles. However, we believe that our independent review process, which resulted in consistent results across multiple and relatively large subsamples, can alleviate this limitation.

## Conclusions

Here, we demonstrate both population- and author-sources contributing toward bias in AI in healthcare. We find that U.S. and Chinese datasets and authors are disproportionately overrepresented in clinical AI. Image-rich specialties (i.e., radiology, pathology, and ophthalmology) are the most prevalent among clinical AI studies published in 2019. Additionally, the authors of these papers are predominantly male researchers with non-clinical backgrounds.

The rapid evolution of AI in healthcare brings unprecedented opportunities for accurate, efficient, and cost-effective diagnosis superseding the accuracy of clinical specialists, but with minimal human input–especially valuable in resource-poor settings where specialist input is relatively scarcer. Although medicine stands to benefit immensely from publicly available anonymised data informing AI-based models, pervasive disparities in global datasets should be addressed. In the long-term, this will require the development of technological infrastructure in data-poor regions (i.e., cloud storage and computer speed). Meanwhile, researchers must be particularly diligent with external validation and re-calibration of their models to elucidate how and when AI/ML findings derived from a local patient cohort can be applied internationally to heterogeneous populations. While investments in technology architecture are being made, these efforts should at least help reduce disparities in access to AI/ML-driven tools.

While clinical AI models trained under certain assumptions and contexts may perform flawlessly among patients of the medical centre at which they were built, they may fail altogether elsewhere. As the scope of influence of AI in healthcare grows, it is critical for both clinical and non-clinical researchers to understand who may be negatively affected by model bias through underrepresentation and to assess the extent of model utility externally in order to make the vast amount of data being collected and its AI applications meaningful for a broader population.

## Author Contributions

**Conceptualization:** Leo Anthony Celi, Jacqueline Cellini, Edward Christopher Dee, William Greig Mitchell, Joseph Paguio, Joel Park.

**Data curation:** Leo Anthony Celi, Edward Christopher Dee, William Greig Mitchell, Lama Moukheiber, Julian Schirmer, Julia Situ, Joseph Paguio, Seth Yao.

**Formal analysis:** Franck Dernoncourt, Rene Eber, William Greig Mitchell, Julian Schirmer, Julia Situ, Joel Park.

**Investigation:** Leo Anthony Celi, Franck Dernoncourt, Rene Eber, William Greig Mitchell, Joel Park.

**Methodology:** Leo Anthony Celi, Edward Christopher Dee, Franck Dernoncourt, Rene Eber, William Greig Mitchell, Joel Park.

**Project administration:** Leo Anthony Celi, Franck Dernoncourt, William Greig Mitchell.

**Resources:** Marie-Laure Charpignon, Edward Christopher Dee, Rene Eber, William Greig Mitchell.

**Software:** Marie-Laure Charpignon, Rene Eber.

**Supervision:** Leo Anthony Celi, Marie-Laure Charpignon, Rene Eber, William Greig Mitchell.

**Validation:** William Greig Mitchell.

**Visualization:** Leo Anthony Celi, Jacqueline Cellini, Marie-Laure Charpignon, Edward Christopher Dee, Rene Eber, William Greig Mitchell, Julia Situ, Joel Park, Seth Yao.

**Writing – original draft:** Edward Christopher Dee, William Greig Mitchell, Joel Park.

**Writing – review & editing:** Leo Anthony Celi, Jacqueline Cellini, Marie-Laure Charpignon, Edward Christopher Dee, Rene Eber, William Greig Mitchell, Lama Moukheiber, Joseph Paguio, Joel Park, Judy Gichoya Wawira, Seth Yao.

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
