## [Decision Letter · Decision Letter 0]

27 Sep 2021

PDIG-D-21-00034

Perpetuating Healthcare Disparities through Bias in Artificial Intelligence – a Global Review

PLOS Digital Health

Dear Dr. Mitchell,

Thank you for submitting your manuscript to PLOS Digital Health. After careful consideration, we feel that it has considerable merit but does not fully meet PLOS Digital Health’s publication criteria as it currently stands. Therefore, we invite you to submit a revised version of the manuscript that addresses the points raised during the review process.

Please review the reviewer comments and add additional and clarifying text, and perhaps a slight change in the title.

We look forward to receiving your revised manuscript.

Kind regards,

Heather Mattie

Academic Editor

PLOS Digital Health

Journal Requirements:

1. Please amend your detailed Financial Disclosure statement. This is published with the article, therefore should be completed in full sentences and contain the exact wording you wish to be published.

i). Please include all sources of funding (financial or material support) for your study. List the grants (with grant number) or organizations (with url) that supported your study, including funding received from your institution. 

ii). State the initials, alongside each funding source, of each author to receive each grant.

iii). State what role the funders took in the study. If the funders had no role in your study, please state: “The funders had no role in study design, data collection and analysis, decision to publish, or preparation of the manuscript.”

iv). If any authors received a salary from any of your funders, please state which authors and which funders.

2. Please update the completed 'Competing Interests' statement, including any COIs declared by your co-authors. If you have no competing interests to declare, please state "The authors have declared that no competing interests exist". Otherwise please declare all competing interests beginning with the statement "I have read the journal's policy and the authors of this manuscript have the following competing interests:"

3. Please provide separate figure files in .tif or .eps format only and remove any figures embedded in your manuscript file.  Please ensure that all files are under our size limit of 20MB.  

For more information about how to convert your figure files please see our guidelines: https://journals.plos.org/digitalhealth/s/figures

Once you've converted your files to .tif or .eps, please also make sure that your figures meet our format requirements

4. Tables should not be uploaded as individual files. Please remove these files and include the tables in your manuscript file.

5. We have noticed that you have cited 'Supplementary Figure X' in the manuscript file. However, there is no corresponding file uploaded to the submission. Please ensure that all files are present to ensure that your paper is fully reviewed.

Additional Editor Comments (if provided):

This is a timely piece and with the addition of text descriptions and clarifications, and perhaps a slight change in the title, will be a great contribution to the literature. In particular additional reflections on the potential gaps in the available methods for detecting bias in medical ML studies such as nationality of authors vs country of current institutional affiliation, and the accuracy of gender based on first names from multiple ethnicities. 

Reviewers' comments:

Reviewer's Responses to Questions

**Comments to the Author**

1. Does this manuscript meet PLOS Digital Health’s publication criteria? Is the manuscript technically sound, and do the data support the conclusions? The manuscript must describe methodologically and ethically rigorous research with conclusions that are appropriately drawn based on the data presented.

Reviewer #1: Yes

Reviewer #2: Partly

Reviewer #3: Yes

2. Has the statistical analysis been performed appropriately and rigorously?

Reviewer #1: Yes

Reviewer #2: N/A

Reviewer #3: N/A

3. Have the authors made all data underlying the findings in their manuscript fully available (please refer to the Data Availability Statement at the start of the manuscript PDF file)?

Reviewer #1: Yes

Reviewer #2: No

Reviewer #3: Yes

4. Is the manuscript presented in an intelligible fashion and written in standard English?

Reviewer #1: Yes

Reviewer #2: Yes

Reviewer #3: Yes

5. Review Comments to the Author

Reviewer #1: PERPETUATING HEALTHCARE DISPARITIES THROUGH BIAS IN ARTIFICIAL

Review

The study authors posit that the disparities in data sources of clinical AI studies done in 2019 and referenced in PubMed—namely, country or countries where the study was done, specialty of researchers, nationality of researchers, sex and expertise of researchers-- lead to bias, which in turn perpetuates “healthcare disparities” in general and specifically disparity in applicability of study findings toward healthcare of the study population (data source of study), vs the healthcare of the general disease population.

The methodology of a typical AI clinical study looks like this:

Clinical question regarding a disease is generated based on a real-life situationhypothesis (H1) is generated �data is gathered by researcher from a study population with said disease the data is split into 70:20:10. 70% of the data is used for model development, 20% for model validation, and 10% for model testing (augmentation sometimes done to balance the data)�training of model is done until performance is above 90%validation dataset is used to refine the model�more and more data is gathered for training and testing until testing performance is higher than 80%The algorithm is tested on a bigger or different population but with the same diseasethe clinical question is considered answered once ROC is .80 or higher in the general disease population—>new knowledge is added to the current pool of knowledge regarding the disease in question.

The sources of bias in AI research methodology that were highlighted in the body of this paper (but not clearly stated in the title) are found in the beginning of any AI clinical study. This is the part of the methodology where humans are involved—in the conception of the AI clinical study design and in the generation of the hypothesis (H1) to be tested. Another source of bias is the creation of the AI model that fails to take into account inherent differences in disease characteristics in human populations from which training, internal validation and testing data is gathered vs. the disease characteristics in human populations where the study findings will be applied.

What may be a more suitable title, based on the methodology and findings of this study, is SOURCES OF BIAS IN ARTIFICIAL INTELLIGENCE CLINICAL STUDIES THAT PERPETUATE DISPARITIES IN HEALTHCARE. This study’s authors could then go on to identify 2 sources of bias in AI-powered studies: 1) the researchers who ask the clinical question and gather and analyze relevant data (authors’ clinical specialty, nationality, sex, expertise may introduce bias), and 2) the population of interest from whom the clinical data is gathered, which may have different disease characteristics from the general disease population upon whom the study findings will be applied (country source). A more thorough review of literature on the existence of disparities in global healthcare would serve as a relevant background to this study.

Regarding limitations of the study. This study uses AI to study the biases inherent in AI clinical studies. This begs the question: do the same biases apply to non-clinical AI studies such as this? Are the researchers of this non-clinical AI study biased because their study was conducted in the United States, because of the nationality (needs to be better defined) of the first and last authors, because of the gender of the first and last authors and because of their expertise? An introspective answer to these questions may further help in proving the point of this study, that all humans are biased, and that these biases affect the direction of healthcare research and application in the real world.

Reviewer #2: Thank you for the opportunity to review your paper on this important topic. The work behind this is considerable.

However, I think it needs a re-think in terms of context. The paper presented extols a single view, and I think needs a more balanced view - in effect the paper risks losing impact because it reads as biased in itself.

Much of the current general research literature is arguably biased or potentially biased - including populations, settings, authors, and institutions. Therefore AI based research needs to be viewed through this lens. It is because of this bias that one of the key tenants of Evidence Based Medicine is the question ‘does this apply to my patient?’. AI research should be viewed through the same lens. By referring back to EBM, it can put AI based research in to a similar context to other research, making it potentially more digestible and more mainstream.

The choice of current institution as representing the world view of the authors is potentially misleading. Looking at the authors of this paper there is broad cultural diversity, not represented in the list of affiliations. Therefore this tool is creating its own bias. This needs to be recognised and discussed. The same may apply to the name API.

There are some developing tools to detect bias in datasets – something that is not represented in any previous non-AI research. Although these are still not often used, in a paper on bias I think their existence should be recognized, and use encouraged.

The growing availability of datasets also needs a balanced view. Noting that there are real and potential biases in these, it is also important to recognize these have already had a valuable contribution to the field, and their ongoing development and use should be encouraged rather than discouraged, noting the limitations. Their availability becomes an attractant to time investment, which potentially improves the opportunity to democratize Healthcare.

The ability to transfer learn and thus customize to local populations needs more attention. This technique allows the power of large datasets to help inform smaller datasets, meaning it is simpler to get a return on investment in lower resource environments, thereby encouraging development in this area. It also leverages the investment in resource rich environments. Also of note is that for most non-AI interventions this opportunity rarely if ever exists – either to repeat the intervention, or tailor the intervention to local population and parameters.

The topic is important. I think a more balanced view, including highlighting the opportunities along with potential fixes as well as the risks, is important to get this message across.

Reviewer #3: It is a very important and interesting manuscipt about global disparities through bias in AI. The authors addressed the importance of AI in medical care and through a rigorous and extensive review and application of transfer-learning techniques. They found factors that could account for disparities in building of AI and its application in medical care.

The authors mention in the abstract that databases came mainly from the USA and China. Looking at the results it is noticeable that database, at least the first 10 places, were from high income countries. It is important to introduce a phrase in abstract indicating such observation, because the authors make emphasis, in the interpretation (abstract) that it is important to develop infrastructure for AI in data-poor regions.

In page 16 the authors speak about the over-representation of radiology, the same argument “access to image-data” applies for pathology, which was the other over-represented specialty.

The authors do not mention as a limitation on gender analysis, that the same names are used in some countries for women and men.

Even though AI is very important in Medicine, physicians do not (at least not in many countries) acquire the basic knowledge of AI when they are medical students. Same happens with specialists. Since the manuscript focuses on disparities, it would be convenient to mention the handicap that medical students have in AI knowledge. That explains why a high proportion of the main AI authors are not clinicians.

Even though I accepted the manuscript I would like the authors to introduce in the abstract that the first ten databases came from high income countries.

6. PLOS authors have the option to publish the peer review history of their article (what does this mean?). If published, this will include your full peer review and any attached files.

**Do you want your identity to be public for this peer review?** For information about this choice, including consent withdrawal, please see our Privacy Policy.

Reviewer #1: **Yes: **BEATRICE TIANGCO

Reviewer #2: No

Reviewer #3: **Yes: **Cleva Villanueva

---

## [Decision Letter · Decision Letter 1]

7 Feb 2022

Sources of Bias in Artificial Intelligence that Perpetuate Healthcare Disparities - a Global Review

PDIG-D-21-00034R1

Dear Dr Mitchell,

We are pleased to inform you that your manuscript 'Sources of Bias in Artificial Intelligence that Perpetuate Healthcare Disparities - a Global Review' has been provisionally accepted for publication in PLOS Digital Health.

Best regards,

Hamish S Fraser, MBCHB MSc

Section Editor

PLOS Digital Health

Thank you for addressing the reviewer comments. We are happy to accept the manuscript.

Reviewer Comments (if any, and for reference):

Reviewer's Responses to Questions

**Comments to the Author**

1. If the authors have adequately addressed your comments raised in a previous round of review and you feel that this manuscript is now acceptable for publication, you may indicate that here to bypass the “Comments to the Author” section, enter your conflict of interest statement in the “Confidential to Editor” section, and submit your "Accept" recommendation.

Reviewer #1: All comments have been addressed

Reviewer #3: All comments have been addressed

2. Does this manuscript meet PLOS Digital Health’s publication criteria? Is the manuscript technically sound, and do the data support the conclusions? The manuscript must describe methodologically and ethically rigorous research with conclusions that are appropriately drawn based on the data presented.

Reviewer #1: Yes

Reviewer #3: Yes

3. Has the statistical analysis been performed appropriately and rigorously?

Reviewer #1: N/A

Reviewer #3: Yes

4. Have the authors made all data underlying the findings in their manuscript fully available (please refer to the Data Availability Statement at the start of the manuscript PDF file)?

Reviewer #1: Yes

Reviewer #3: Yes

5. Is the manuscript presented in an intelligible fashion and written in standard English?

Reviewer #1: Yes

Reviewer #3: Yes

6. Review Comments to the Author

Reviewer #1: No further comments

Reviewer #3: This reviewer has carefully read and analyzed the second submission of the manuscript 00034R to PLOS Digital Health. The manuscript addresses an important issue in modern health care, Artificial Intelligence and bias that affect equality in healthcare.

The authors have properly answered the questions of the reviewers and have changed the manuscript accordingly. The manuscript is understandable, shows important data and points out subjects that should change in the field of application of AI on medicine and healthcare. This reviewer only noticed one word that would be convenient to change. The word “sex” could be changed by “gender”. The opinion of this reviewer is that the manuscript meets all the criteria to be published in PLOS Digital Health.

7. PLOS authors have the option to publish the peer review history of their article (what does this mean?). If published, this will include your full peer review and any attached files.

**Do you want your identity to be public for this peer review?** For information about this choice, including consent withdrawal, please see our Privacy Policy.

Reviewer #1: **Yes: **Beatrice J. Tiangco

Reviewer #3: **Yes: **Cleva Villanueva
